# The Effects of Obstructive Sleep Apnea on the Cardiovascular System: A Comprehensive Review

**DOI:** 10.3390/jcm13113223

**Published:** 2024-05-30

**Authors:** Michael V. DiCaro, KaChon Lei, Brianna Yee, Tahir Tak

**Affiliations:** Department of Medicine, Kirk Kerkorian School of Medicine at UNLV, Las Vegas, NV 89102, USA; michael.dicaro@unlv.edu (M.V.D.); kachon.lei@unlv.edu (K.L.); brianna.yee@unlv.edu (B.Y.)

**Keywords:** sleep apnea, CPAP therapy, atrial fibrillation, heart failure, hypertension, stroke

## Abstract

Obstructive sleep apnea (OSA) is an increasingly relevant cause of cardiovascular morbidity worldwide. Although the association between OSA and the cardiovascular system is well-known, the extent of its effects is still a topic of interest, including pathophysiologic mechanisms, cardiovascular sequelae, and OSA therapies and their effects. Commonly described mechanisms of cardiovascular etiologies revolve around sympathetic activation, inflammation, and intermittent hypoxia resulting from OSA. Ultimately, these effects lead to manifestations in the cardiovascular system, such as arrhythmias, hypertension, and heart failure, among others. The resulting sequelae of OSA may also have differential effects based on gender and age; several studies suggest female gender to have more susceptibility to cardiovascular mortality, as well as an increase in age. Furthermore, several therapies for OSA, both established and emerging, show a reduction in cardiovascular morbidity and may even reduce cardiovascular burden. Namely, the establishment of CPAP has led to improvement in hypertension and cardiac function in patients with heart failure and even reduced the progression of early stages of atherosclerosis. Effective management of OSA decreases abnormal neural sympathetic activity, which results in better rhythm control and blood pressure control, both in waking and sleep cycles. With newer therapies for OSA, its effects on the cardiovascular system may be significantly reduced or even reversed after long-term management. The vast extent of OSA on the cardiovascular system, as well as current and future therapeutic strategies, will be described in detail in this review.

## 1. Introduction

Obstructive sleep apnea (OSA) is a highly prevalent sleep disorder characterized by repeated episodes of upper airway obstruction during sleep, leading to intermittent cessation or reduction in airflow and disrupted breathing patterns. The condition is estimated to affect nearly 1 billion people worldwide, associated with significant morbidity and mortality [1]. A multi-ethnic study examining sleep-disordered breathing found the prevalence of OSA to be 62.9% in white individuals, 63.5% in African American individuals, 71.5% in Hispanic individuals, and 66.4% in Chinese individuals [2]. Furthermore, many studies have shown an increased prevalence of OSA in males compared to females [2,3,4,5]. One study estimated the prevalence to be two-fold greater in men [6]. Additionally, untreated severe OSA in women was noted to have increased cardiovascular mortality as compared to men [7].

While OSA is primarily recognized as a respiratory disorder, accumulating evidence suggests that it exerts profound effects on the cardiovascular system, contributing to a spectrum of cardiovascular abnormalities and increasing the risk of adverse cardiovascular events [8,9]. Individuals with OSA often present with comorbidities such as hypertension, coronary artery disease, atrial fibrillation, heart failure, and stroke, highlighting the broad impact of OSA on cardiovascular health [8,10]. Moreover, emerging evidence suggests that OSA may coexist with cardiovascular conditions and contribute to their pathogenesis and progression through various mechanisms [9].

Understanding the complex interplay between OSA and cardiovascular disease is crucial for clinicians and researchers alike, as it has important implications for risk stratification, diagnosis, management, and outcomes of patients with OSA. Elucidating the underlying mechanisms linking OSA to cardiovascular pathology may provide insights into novel therapeutic targets and interventions to mitigate the cardiovascular burden associated with OSA.

In this review, we aim to provide a comprehensive overview of the effects of OSA on the cardiovascular system. We will examine the epidemiology of OSA and its association with various cardiovascular conditions, explore the underlying pathophysiological mechanisms linking OSA to cardiovascular disease, discuss the clinical implications of OSA-related cardiovascular complications, and review current approaches to diagnosing and managing cardiovascular comorbidities in individuals with OSA. By synthesizing existing evidence and highlighting areas of ongoing research, this review seeks to deepen our understanding of the intricate relationship between OSA and cardiovascular health and guide future directions for clinical practice and research in this field.

## 2. Methods

Bibliographic databases, including PubMed and Embase, were used for this literature review to answer the question, “What are the cardiovascular consequences of obstructive sleep apnea?” Keywords used in this search included obstructive sleep apnea, OSA, sleep-disordered breathing, cardiovascular effects, cardiac effects, cardiovascular consequences, and cardiac consequences. Article types included observational studies, epidemiological studies, retrospective studies, randomized controlled trials, systematic reviews, and meta-analyses. Several articles were obtained from previously referenced publications.

## 3. Mechanisms of Action of OSA (Summarized in Figure 1)

The mechanism underlying OSA is complex and multifactorial, involving a combination of anatomical, physiological, and neurological factors that interact to produce airway collapse and subsequent respiratory disturbances during sleep. Anatomical abnormalities, reduced muscle tone, negative intrathoracic pressure, intermittent hypoxia, sympathetic activation, and alterations in neural control mechanisms all play integral roles in the mechanism of OSA. Understanding these mechanisms is essential for developing targeted diagnostic and therapeutic strategies to mitigate airway obstruction and improve sleep quality and cardiovascular health in individuals with OSA.

**Figure 1 jcm-13-03223-f001:**
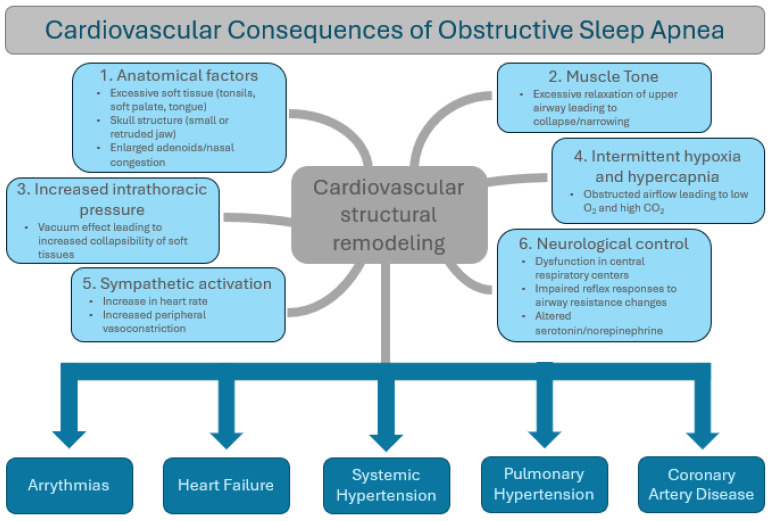
Pathophysiologic mechanisms of obstructive sleep apnea on the cardiovascular system.

### 3.1. Anatomical Factors

One of the primary contributors to OSA is anatomical abnormalities or alterations in the upper airway structure. These may include excessive soft tissue in the throat, such as enlarged tonsils, a long soft palate, or a large tongue, narrowing the airway and predisposing it to collapse during sleep [11,12,13]. Additionally, abnormalities in the skeletal structure of the face and neck, such as a small or retruded jaw (micrognathia or retrognathia), can further compromise airway patency by reducing the space available for airflow [14]. Enlarged adenoids or nasal congestion due to allergies or nasal septum deviation can also contribute to airway obstruction by obstructing the nasal passages and increasing resistance to airflow [11,12].

### 3.2. Muscle Tone

During sleep, the upper airway muscles—including the tongue, soft palate, and throat muscles—naturally relax to some extent. However, in individuals with OSA, this relaxation may be excessive or insufficiently counteracted by muscle tone, leading to collapse or narrowing of the airway [15,16]. Factors such as obesity, alcohol consumption, sedative medications, and smoking can further exacerbate muscle relaxation and increase the likelihood of airway collapse [15,16,17,18]. Reduced muscle tone in the upper airway also impairs the ability of these muscles to maintain airway patency in the presence of negative intrathoracic pressure generated during breathing efforts.

### 3.3. Negative Intrathoracic Pressure

When an individual attempts to breathe against a collapsed or obstructed upper airway, negative pressure is generated within the chest cavity (intrathoracic pressure). This negative pressure can exacerbate airway collapse by increasing the collapsibility of the soft tissues surrounding the airway. During inspiration, the diaphragm contracts, and the chest wall expands, creating a vacuum effect that can further draw collapsed or flaccid soft tissues inward, contributing to airway obstruction. This phenomenon is particularly pronounced during obstructive events in OSA and can worsen the severity of respiratory disturbances [15].

### 3.4. Intermittent Hypoxia and Hypercapnia

As airflow is obstructed during apnea episodes, oxygen levels in the blood decrease, leading to hypoxemia and hypercapnia. Intermittent hypoxia and hypercapnia trigger physiological responses aimed at restoring normal oxygenation and ventilation. These responses include sympathetic nervous system activation, increased heart rate, and peripheral vasoconstriction, increasing the drive to breathe and overcoming airway obstruction. However, these compensatory mechanisms are often inadequate in fully restoring airflow, leading to recurrent cycles of hypoxia, hypercapnia, and respiratory efforts interrupted by apnea and arousal from sleep [15,19].

### 3.5. Sympathetic Activation

OSA increases sympathetic nervous system activity, particularly during apnea episodes and subsequent arousals from sleep [15]. Sympathetic activation leads to elevated heart rate, blood pressure, and peripheral vasoconstriction, which can exacerbate cardiovascular strain and contribute to the pathophysiology of OSA-related cardiovascular complications [15,20]. The surge in sympathetic tone during apnea events may further destabilize the upper airway and promote breathing efforts in an attempt to overcome airway obstruction.

### 3.6. Neurological Control of Upper Airway

The control of upper airway patency during sleep is complex and involves a delicate balance of neural inputs from various regions of the brainstem. Disruptions in this neural control mechanism, such as dysfunction in the central respiratory centers or impaired reflex responses to changes in airway resistance, can predispose individuals to airway collapse and contribute to the pathogenesis of OSA [15]. Alterations in the neurotransmitter systems involved in upper airway control, such as serotonin and norepinephrine, may also play a role in modulating airway muscle tone and responsiveness to respiratory stimuli [21].

## 4. Diagnosis

Diagnosing OSA typically involves a combination of clinical evaluation, sleep studies, and assessment of symptoms and risk factors. Several diagnostic and assessment tools are commonly used to evaluate individuals suspected of having OSA.

### 4.1. Clinical History and Physical Examination

A comprehensive clinical history and physical examination are essential components of the diagnostic evaluation for OSA. Healthcare providers assess symptoms such as snoring, witnessed apneas, daytime sleepiness, morning headaches, and nocturnal choking or gasping. They also inquire about risk factors for OSA, including obesity, neck circumference, family history, and medical conditions such as hypertension and diabetes.

### 4.2. Sleep Questionnaires

Various validated questionnaires are available to assess the severity of OSA symptoms and daytime sleepiness. The Epworth Sleepiness Scale (ESS) [22] is commonly used to quantify daytime sleepiness, while other questionnaires, such as the Berlin Questionnaire [23] and STOP-BANG questionnaire [24], assess OSA risk based on symptoms and clinical features [25].

### 4.3. Polysomnography (PSG)

Polysomnography is the gold standard for diagnosing OSA. In cases of a negative, inadequate, or inconclusive home sleep apnea test (see section below), PSG may be done for diagnosis of OSA [26]. PSG is a comprehensive sleep study conducted in a specialized sleep laboratory or home environment [26]. It involves monitoring various physiological parameters during sleep, including airflow, respiratory effort, oxygen saturation, heart rate, and sleep stages [16]. PSG provides detailed information about the frequency and severity of apnea and hypopnea events and associated sleep disruptions and physiological abnormalities.

### 4.4. Home Sleep Apnea Testing (HSAT)

Home sleep apnea testing is an alternative to in-laboratory PSG for diagnosing uncomplicated OSA in select patients [25,26]. HSAT typically involves the use of portable monitoring devices worn at home to record basic physiological parameters such as airflow, respiratory effort, and oxygen saturation. While HSAT is more convenient and cost-effective than PSG, it may not capture all aspects of sleep architecture and may be less accurate in detecting milder forms of OSA or comorbid sleep disorders [25,26]. Additionally, patients with comorbidities including significant cardiac or respiratory disease, neuromuscular conditions that may affect respiratory muscle strength, chronic opioid use, awake hypoventilation, history of stroke, or severe insomnia are not recommended to utilize HSAT and should undergo PSG instead [26]. 

### 4.5. Apnea–Hypopnea Index (AHI)

AHI is a key parameter used in the diagnosis and severity assessment of OSA. It quantifies the frequency and severity of apnea and hypopnea events per hour of sleep. While it has been challenged in recent years, it is still regarded as an important tool to assess the severity of OSA. The significance of the AHI lies in its ability to provide a standardized measure of the severity of OSA and guide clinical decision-making in the management of this sleep disorder. Based on the AHI, OSA can be classified into different severity categories, ranging from mild to severe. The severity classifications are as follows [25]:Mild OSA: AHI 5–14 events per hourModerate OSA: AHI 15–29 events per hourSevere OSA: AHI ≥ 30 events per hour

These severity categories help clinicians and patients understand the impact of OSA on sleep quality and health outcomes and guide treatment decisions. Table 1 depicts the AHI scoring system.

## 5. Prevalence of Cardiovascular Effects

The effects of OSA on the cardiovascular system are numerous and well-documented. Untreated OSA affects nearly every element of the cardiovascular system and is deemed an independent risk factor for numerous forms of cardiovascular disease (CVD), including heart arrhythmias such as atrial fibrillation and ventricular tachycardia, vascular diseases such as coronary artery disease, stroke, and hypertension, and heart failure [5,10]. Not only do adults with OSA have an increased risk of developing CVD, but they also have significantly worse outcomes related to CVD [6,10]. The cardiovascular consequences of OSA are depicted in Figure 1. Data regarding the cardiovascular effects of OSA are described below (summarized in Table 2).

### 5.1. Cardiac Arrhythmias

OSA has been associated with numerous cardiac arrhythmias, including atrial arrhythmias, ventricular arrhythmias, heart block, sinus arrest, ectopies, and increased incidence of heart rate variability. About 30–50% of patients with OSA are also diagnosed with cardiac arrhythmias [27]. Several of the most prevalent associations are described in detail:Atrial fibrillation: OSA is a well-established independent risk factor for the development of atrial fibrillation (AF). The prevalence of AF in the general population is roughly 1–2% [28]. In patients with OSA, the prevalence is approximately 5% [29]. Growing evidence also implicates a concomitant prevalence of AF of 21–74% in patients with OSA, which may suggest that OSA may play a role in AF pathogenesis and development and may also trigger the persistence of AF [30,31]. Numerous clinical trials and meta-analyses have confirmed the strong association between OSA and AF, including one meta-analysis performed in 2018 consisting of 12 studies and nearly 20,000 patients [31]. OSA severity correlates with a higher incidence of AF and may predict AF recurrence after cardioversion and/or with ablation procedures. Furthermore, standard antiarrhythmic therapies in patients with severe OSA are much less likely to be successful compared to patients without OSA [32]. The presence of OSA is associated with an increased incidence of AF in patients with heart failure, coronary artery disease, and hypertrophic cardiomyopathy as well [33,34].Sinus arrest: The prevalence of sinus arrest in patients with OSA has been documented to be as high as 11%, with the majority of cases lasting between 2.5 and 12 s [35]. Other studies using polysomnography have documented sinus arrests lasting greater than 2 s in 4% of healthy study participants without known cardiac abnormalities or OSA [36]. According to the European multicenter polysomnographic study, 58% of patients with implanted pacemakers for sinus node dysfunction had symptomatic OSA [37]. The pathophysiologic mechanisms underlying OSA likely create an environment conducive to sinus arrest. The long-term structural heart changes occurring in the left atrium are also likely involved. Despite this, OSA has also been shown to cause nocturnal bradyarrhythmias and sinus arrest in the acute nocturnal setting in the absence of cardiac conduction disease [35,38].Bradyarrhythmias: These occur in up to 18% of patients with OSA [35]. The most frequently observed bradyarrhythmias are sinus bradycardia, sinus arrest leading to bradycardia (described above), and atrioventricular (AV) nodal block (predominantly grade II and III). In comparison, the prevalence of nocturnal bradyarrhythmias in healthy adults is roughly 3% [39]. Several studies have reported the incidence of grade II and III AV nodal block to occur in about 10% of patients with OSA, compared with 1% in the healthy elderly population [35,39,40]. In the European multicenter polysomnography study, 68% of patients treated with a pacemaker for atrioventricular block had minimally symptomatic OSA, and 27% fulfilled the criteria for severe OSA. Similarly to other arrhythmias associated with OSA, the incidence of bradycardia arrhythmias in patients with OSA is likely correlated with the degree of severity. Significantly higher incidences of bradyarrhythmias have been observed in OSA patients with higher AHI [37,40,41].Ventricular repolarization abnormalities and arrhythmias: QTc interval dispersion and prolongation and ventricular arrhythmias are seen more frequently in patients with OSA compared to the general population [10]. In particular, the QTc interval has been shown to be prolonged during the onset of apnea (482 +/− 34 msec) with the return to baseline during apnea and the post-apnea hyperventilation period [42]. Both monomorphic and polymorphic ventricular tachycardias (VTs) have been reported in higher incidence in patients with OSA [43]. In patients with co-existing heart failure and OSA, OSA was found to be an independent risk factor for ventricular arrhythmias [44]. Furthermore, patients with OSA undergoing catheter ablation for ventricular arrhythmias are associated with a higher rate of recurrence when compared to the general population [45]. The Sleep Heart Health Study, a population study involving 566 patients, showed an increased prevalence of complex ventricular ectopy (ventricular bigeminy, trigeminy, and quadrigeminy) when compared to the general population [29].

### 5.2. Heart Failure

The Sleep Heart Health Study reported a 58% higher adjusted risk of incident heart failure in patients with severe OSA [46,47]. Other studies have reported similar findings, showing an elevated incidence and prevalence of sleep-disordered breathing and OSA in patients with congestive heart failure [47]. In addition to acting as a risk factor for the development and progression of heart failure, patients with OSA and co-existing heart failure tend to have worse outcomes and increased cardiovascular events compared to the general population [47]. In this study, cardiovascular events were defined as hospitalization for congestive heart failure, cardiovascular death, or acute coronary syndrome [7,47]. In the general population, the prevalence of OSA with an AHI exceeding 10–15 is roughly 7–10% [4]. This is significantly lower compared to patients with heart failure and coexisting OSA, who demonstrate an AHI of 11–53% [48,49,50]. Overall, clinicians should consider the significance of OSA on the development, progression, and elevated risk of complications related to heart failure.

### 5.3. Systemic Hypertension

OSA has been considered a risk factor for hypertension since the 1980s when several studies associated systemic hypertension with snoring and nighttime apnea [51,52]. A large cross-sectional study involving 6132 subjects revealed a strong association between systemic hypertension, sleep-disordered breathing, and sleep apnea in middle-aged and older individuals of different sexes and ethnic backgrounds. Even when accounting for possible confounding variables such as BMI, neck circumference, age, alcohol intake, and smoking, mean systolic and diastolic blood pressure and the presence of hypertension increased significantly with increasing AHI (at least an AHI of 30/h) [53].

### 5.4. Pulmonary Hypertension (PH)

Studies have reported a prevalence of PH ranging from approximately 17% to as high as 80% among individuals with moderate to severe OSA, depending on diagnostic methods [54]. These prevalence estimates suggest that PH is a relatively common comorbidity in OSA and may have important patient management and outcomes implications.

### 5.5. Coronary Artery Disease (CAD)

Multiple observational studies and meta-analyses have demonstrated a higher prevalence of CAD among individuals with OSA compared to those without OSA. These studies have reported varying prevalence rates of CAD in individuals with OSA, ranging from approximately 20% to 60%, depending on factors such as the severity of OSA, the presence of comorbidities, and the characteristics of the study population [55]. Patients with OSA and coexisting coronary artery disease have also been shown to have a greater percentage of new myocardial infarctions, restenosis after percutaneous coronary intervention, and cardiovascular death [56,57]. The benefit of OSA treatment in CAD is clear, shown by several studies highlighting a decrease in the occurrence of new cardiovascular events [47].

### 5.6. Stroke

OSA is a risk factor for cerebrovascular disease and stroke, and the risk of OSA following stroke is also reportedly higher compared to the general population [58,59,60]. The Sleep Heart Health Study also revealed greater odds for stroke in the highest AHI quartile when compared to the lowest quartile, indicating that the severity of OSA is also associated with a higher risk of stroke. The risk of stroke in patients with severe untreated OSA (AHI > 30) may be highest in the elderly population aged 70–100 years old, as one 6-year longitudinal study revealed 2.5 times increased risk of stroke compared to patients with AHI less than 30. One observational cohort study enrolling 1022 patients noted a significant association between OSA and a combined rate of transient ischemic attack, stroke, and all-cause mortality [61].

**Table 2 jcm-13-03223-t002:** Summarization table of cardiovascular consequences of OSA including epidemiology and pathophysiology.

Cardiovascular Condition	Prevalence/Incidence among OSA Patients	Prevalence/Incidence in General Population	Pathophysiology
Cardiac arrhythmias Atrial fibrillation Sinus arrest Bradyarrhythmia Ventricular arrhythmia	30–50% [27] 4.8% [29] 11% [35] 18% [35] 25% [29]	1.5–5% [62] 0.9% [29] 4% [36] 3% [39] 14.5% [29]	Increased intrathoracic pressure Intermittent hypoxia/hypercapnia Sympathetic activation
Heart failure Systolic heart failure Diastolic heart failure	11–37% [63] 56% [63] 62% [64]	4.4% [65] 20% [63] 2.9% [65]	Intermittent hypoxia/hypercapnia Sympathetic activation Neurologic control
Systemic hypertension	59% [53]	21% [53]	Intermittent hypoxia/hypercapnia Sympathetic activation Neurologic control
Pulmonary hypertension	27–30% [66]	13–28% [67]	Increased intrathoracic pressure Intermittent hypoxia/hypercapnia Neurologic control
Vascular disease Coronary artery disease Myocardial ischemia	16.6% [63] 30% [68] 38.4 [69]	13% [63] 12% [68] 26.6 [69]	Intermittent hypoxia/hypercapnia Neurologic control

## 6. Mechanism of Action of Cardiovascular Effects

The pathophysiologic mechanisms of OSA are well-known to cause significant deleterious effects on the cardiovascular system. Potential causative factors are numerous and are outlined below.

### 6.1. Cardiac Arrhythmias

In the short term, OSA can significantly impact heart rate and rhythm, particularly during episodes of apnea (breathing pauses) and subsequent arousal from sleep. OSA has been identified as a risk factor for many types of arrhythmia, primarily bradycardias [70,71]. OSA can lead to dynamic fluctuations in heart rate during sleep, characterized by increases during apnea episodes and arousals, as well as potential episodes of bradycardia.

These acute episodes, primarily triggered by the sympathetic and parasympathetic nervous systems, have overlapping pathophysiologic mechanisms. They are described in detail below:Bradycardia: Upper airway obstruction elicits the “diving reflex”, which manifests physiologically as increased sympathetic vasoconstriction to muscles and viscera in order to maintain tissue perfusion. The lack of lung expansion from upper airway obstruction prevents the stretching of vagolytic fibers in the lungs, causing a rise in blood pressure and vagally induced reflex bradycardia [9]. This frequently occurs during apneic episodes. Additionally, the decrease in intrathoracic pressure during apnea leads to a transient increase in venous return to the heart and filling of the right atrium. The baroreceptors sensed this increase in cardiac preload, leading to increased parasympathetic afferent activity and reduced heart rate [32,72]. As bradycardia can be observed in healthy individuals during sleep, the bradycardia is generally more pronounced and significant in individuals with OSA during apneic episodes [73].Heart Rate Variability: In prolonged states of hypoxemia and hypercapnia, cardiac output usually increases in a compensatory fashion to maintain tissue perfusion. As such, the heart rate may increase in patients with OSA after an apneic episode. As the apnea episode ends and the individual partially or fully awakens to resume breathing, there is often a surge in sympathetic nervous system activity. This arousal response can lead to further increases in heart rate, contributing to fluctuations in heart rate variability throughout the night. Additionally, when parasympathetic tone predominates, heart rate slows, and bradycardia is observed, as described above. The converse reaction occurs when a sympathetic tone predominates and causes the heart rate to rise. The autonomic nervous system therefore controls heart rate variability during sleep in patients with OSA [32].First Degree Heart Block and Sinus Arrest: Prolonged PR interval and prolonged sinus arrest occur in patients with OSA to a greater degree than in the general population [32]. Similarly to the mechanisms of underlying bradycardia in OSA patients, these rhythm variations occur primarily because of the autonomic effects associated with prolonged apnea, oxygen desaturation, changes in cardiac hemodynamics, and variations in intrathoracic pressure [32,72].Premature Ventricular Complexes (PVCs): Patients with OSA have been shown to experience a higher percentage of ventricular ectopy, including frequent PVCs and significant PVC burden during level 3 polysomnography compared to the general population [74]. This includes increased incidence of bigeminy and trigeminy [74]. Increased incidence of ventricular ectopy in patients with OSA is likely due to metabolic abnormalities, prolonged hypoxia, and mechanical stress, leading to cardiac remodeling. These remodeling changes may include fibrosis, hypertrophy, and alterations in the electrophysiological properties of cardiac myocytes, which increase the susceptibility to PVCs and other arrhythmias.

Prolonged untreated OSA can cause more significant long-term deleterious effects on the heart’s conduction system. (A lot of this may be a repetition from above.)

Atrial Fibrillation (AF): Intermittent hypoxia due to obstruction during apnea, sympathetic activation, and left atrial remodeling all contribute to the pathogenesis of AF by promoting atrial remodeling, fibrosis, and electrical instability [75]. Additionally, OSA may impair the effectiveness of treatments for AF, which will be further discussed below.Atrioventricular (AV) Block: While OSA itself may not directly cause heart block, the physiological disturbances associated with OSA can create an environment conducive to the development, or exacerbation, of AV block, primarily Mobitz type 2 second-degree block [36]. Chronic exposure to OSA-related stressors can lead to structural and functional changes in the cardiovascular system [7]. These changes may include fibrosis, hypertrophy, and alterations in the architecture of the cardiac conduction system [76]. Over time, cardiac remodeling can disrupt normal electrical conduction and increase the susceptibility to heart block. Individuals with OSA who experience symptoms suggestive of heart block during waking hours, such as dizziness, fainting, or palpitations, should undergo appropriate cardiac evaluation to determine the need for a permanent pacemaker.Ventricular Tachycardia (VT): The interplay of sympathetic activation, intermittent hypoxia, cardiac remodeling, myocardial ischemia, electrolyte imbalance, autonomic dysfunction, and underlying cardiovascular disease creates a conducive environment for the initiation and perpetuation of ventricular tachycardia in individuals with OSA. Chronic OSA-related changes in ventilation and gas exchange may lead to electrolyte imbalances, promoting QTc prolongation and predisposition to the development of polymorphic ventricular tachycardia [42,77]. Specifically, hypokalemia and hypomagnesemia have been associated with chronic untreated OSA [42,78]. Significant cardiac chamber remodeling also occurs due to prolonged, untreated OSA. Unregulated fluxes in parasympathetic and sympathetic activity, fluctuations in intrathoracic pressure, and increased respiratory effort impose mechanical stress on the heart. The repetitive stretching of the myocardium leads to myocardial remodeling and scarring, which in turn contributes to electrical system remodeling and increases the risk of monomorphic and polymorphic VT [79].Sudden Cardiac Death: As a result of prolonged exposure to hypoxia, autonomic dysfunction, and hemodynamic stresses caused by OSA, patients may have a heightened risk of sudden cardiac death, both during nocturnal apneic episodes and during waking hours [80]. Increased intrathoracic pressure causes increased ventricular afterload, which leads to increased myocardial demand, subendocardial ischemia, hypertrophy and fibrosis, and predisposition to malignant arrhythmias. As mentioned previously, hypoxia and electrolyte abnormalities also cause prolonged QTc and an increased propensity for arrhythmogenicity [80,81]. Each of these factors may contribute to the higher incidence of sudden cardiac death in patients with OSA compared to the general population.

### 6.2. Hemodynamic Consequences

The hemodynamic consequences of OSA are multifaceted and include the development and progression of systemic hypertension, heart failure, pulmonary hypertension, and myocardial ischemia. These consequences occur through an interplay between several pathophysiologic mechanisms, including autonomic dysregulation, intrathoracic pressure irregularities, endothelial dysfunction, and alterations in cardiac functional parameters. These disturbances in cardiovascular physiology contribute to cardiovascular strain and increase the risk of adverse cardiovascular outcomes. The hemodynamic consequences of OSA are described in greater detail below:Systemic Hypertension: Intermittent hypoxia and hypercapnia due to recurrent apnea cause elevated sympathetic nervous system response and lead to catecholamine release and increased vascular tone, resulting in systemic hypertension. The degree of hypertension is likely related to the severity of the OSA [15]. OSA also negatively affects sleep efficiency, which is known to be correlated with risk factors for hypertension, including endothelial dysfunction, arterial stiffness, and increased sympathetic activity [32]. Furthermore, obesity is often co-existent with OSA and is also associated with systemic hypertension, though studies have shown that OSA is related to hypertension when adjusting for body mass index [32]. The renin–angiotensin system (RAS) is known to be activated by obesity and has also been shown to be activated by OSA [82].Right Ventricular (RV) Failure: Numerous overlapping pathophysiologic mechanisms are responsible for the development and progression of RV failure in patients with OSA. During inspiration through a narrowed, occluded upper airway, increasingly negative intrathoracic pressure results in blood being drawn into the thorax, augmenting RV preload [9]. Simultaneously, apnea-induced hypoxia causes pulmonary vasoconstriction. Known as the von Euler–Liljesand mechanism, hypoxic pulmonary vasoconstriction causes pulmonary hypertension and increased RV afterload, which causes right heart failure over time [83]. Intermittent hypoxia can also cause oxygen-free-radical production. Reactive oxygen species are known to activate inflammatory pathways that impair vascular endothelial function and can lead to impaired pulmonary vascular vasodilation [84]. Furthermore, OSA causes left ventricular failure (described below) and worsens right ventricular failure.Left Ventricular (LV) Failure: Similar to right ventricular failure, the mechanisms leading to LV failure in OSA are overlapping and multifactorial. Hypertension, RAS activation, obesity, intermittent hypoxia and hypercapnia, and autonomic dysregulation all contribute to LV remodeling. Hypoxic pulmonary vasoconstriction, as described previously, leads to increased RV pressure which causes RV distension, leading to a leftward shift of the interventricular septum during both systole and diastole, which impedes LV filling and results in reduced stroke volume [85]. Intermittent hypoxia and hypercapnia also elicit fluctuations in sympathetic–parasympathetic activity. During sympathetically mediated systemic increases in vascular tone, vasoconstriction occurs, resulting in increased LV afterload [86]. Increased afterload leads to reduced stroke volume and increased end-systolic volume, leading to maladaptive responses by the left ventricle over time, causing concentric hypertrophy and reduced chamber diameter. LV hypertrophy reduces chamber size, causing heart failure with preserved ejection fraction (HFpEF), which eventually deteriorates into heart failure with reduced ejection fraction (HFrEF) [87]. The RAS system is also activated by enhanced sympathetic activity, which causes increased angiotensin II, aldosterone, and subsequent fluid retention, further exacerbating heart failure. Finally, negative intrathoracic pressure also leads to increased atrial remodeling (facilitating atrial fibrillation), elevated LV end-diastolic pressure, increased afterload, and increased cardiac workload leading to increased myocardial oxygen demand [49].Pulmonary Hypertension: As previously mentioned, negative intrathoracic pressure leads to hypoxic pulmonary vasoconstriction, which causes increased RV afterload. Pulmonary vasoconstriction also raises pulmonary artery pressure in the short- and long term. Alveolar hypoxia also causes endothelial remodeling via the production of reactive oxygen species and other vasoactive substances such as serotonin, endothelin, and Rho-kinase. Nitric oxide scavenging via reactive oxygen species also occurs, leading to a reduction in vasodilation. OSA also leads to pulmonary hypertension via hydrostatic mechanisms due to increases in left atrial pressure causing pulmonary venous hypertension and pulmonary interstitial edema. Each of these pathways causes pulmonary vascular remodeling via the proliferation of smooth muscle cells and asymmetric neointimal hyperplasia [66,88].Myocardial Ischemia: Oscillations in blood pressure and cardiac output during apneic episodes lead to increased myocardial oxygen consumption, therapy facilitating type 2 myocardial infarction via supply/demand mismatch [89]. Elevated oxidative stress from hypoxia results in the production of reactive oxygen species, which leads to coronary artery remodeling over time. Abnormal peroxidation of lipids due to increased oxidative burden leads to increased concentration of low-density lipoprotein (LDL), which in turn promotes the formation of coronary arterial vulnerable plaques [90]. Furthermore, prolonged oxidative stress results in the production of oxygen-free radicals and inflammatory mediators. Chronic inflammation, which occurs via oxidative stress and increased vascular tone, ultimately results in atherosclerosis and coronary artery disease over the long term, which further predisposes patients to acute coronary syndrome [91].

### 6.3. Vascular Disease

Extensive effects of OSA on oxidative stress, inflammation, sympathetic activation, endothelial dysfunction, insulin resistance, metabolic dysfunction, and hemodynamic instability create a conducive environment for the development and progression of vascular disease. Specifically, the risk of coronary artery disease and stroke is significantly elevated in patients with OSA. The mechanisms behind these associations are described in detail below:Coronary Artery Disease (CAD): Untreated OSA is associated with a significantly elevated risk of CAD [9]. Untreated OSA is characterized by chronic low-grade inflammation, with elevated levels of inflammatory markers such as C-reactive protein (CRP), interleukin-6 (IL-6), and tumor necrosis factor-alpha (TNF-alpha). Systemic inflammation promotes endothelial dysfunction, oxidative stress, atherosclerosis, and subsequent acute coronary syndrome. The development of CAD via chronic inflammation occurs via redox reactions involving oxidized LDL leading to atheroma formation. As previously described, oxidative stress results in free radical production and subsequent recruitment of inflammatory cells, including neutrophils and monocytes [92,93]. Prolonged exposure to these inflammatory mediators results in endothelial dysfunction and unstable plaque formation. Hypertension, which is highly associated with OSA, also causes CAD via increased vessel stiffness and reduced vascular compliance. These factors, when cumulatively accounted for, contribute to a significant risk in the development of CAD [94].Cerebral Vascular Disease and Stroke: OSA may be an independent risk factor for ischemic stroke [95]. In OSA patients, significant contributing factors include cerebrovascular disease, atrial fibrillation, and sleep fragmentation. Elevated oxidative stress results in endothelial dysfunction, inflammation, and vascular hardening, thus predisposing patients to cerebral ischemia. Cerebral vasculatures are most susceptible to plaque development and ischemia [95]. Atrial fibrillation is also a significant risk factor for ischemic stroke, which is well-documented. Atrial fibrillation in OSA patients is also more difficult to control than in the general population, which leads to an enhanced risk of atrial-fibrillation-induced clot formation (most commonly in the left atrial appendage) and embolization of the brain. OSA also disrupts normal sleep architecture, including rapid eye movement (REM) sleep, which plays a crucial role in cerebral hemodynamics and neurovascular regulation. Fragmented REM sleep may impair cerebral blood flow regulation and increase the risk of stroke [95,96].Peripheral Arterial Disease: Similar mechanisms leading to cerebrovascular disease and coronary artery disease also contribute to the development of peripheral arterial disease in patients with OSA. These mechanisms are marked by repetitive occurrences of hypoxemia and hypercapnia, which promote highly inflammatory environments and oxidative stress, thereby leading to endothelial dysfunction and vascular remodeling [97].

## 7. Treatment Mechanisms and Evidence

The primary treatment for OSA is continuous positive airway pressure (CPAP) therapy. CPAP therapy involves using a machine that delivers a constant flow of air through a mask worn over the nose or both the nose and mouth during sleep. Most complications from obstructive sleep apnea can be prevented with proper CPAP use and compliance. Other therapeutic strategies are aimed at treating the complications of OSA, such as control of arrhythmias, heart failure, obesity, and hypertension. The therapeutic strategies for OSA are discussed below, with a primary focus on CPAP therapy.

### 7.1. CPAP Therapy—How It Works

Maintaining Airway Patency: The continuous flow of pressurized air delivered by the CPAP machine creates a pneumatic splint that keeps the soft tissues of the upper airway—from the back of the throat to the nasal passages—open and prevents them from collapsing or obstructing the flow of air [98]. By maintaining airway patency, CPAP therapy effectively eliminates or reduces apnea episodes and hypopneas, restoring normal breathing patterns during sleep.Preventing Airway Collapse: Individuals with OSA experience repeated episodes of airway collapse or narrowing during sleep, leading to breathing pauses and disruptions in oxygen supply to the body. CPAP therapy counteracts this by delivering constant positive pressure to the airway, preventing it from collapsing or narrowing even during the relaxation of the throat muscles [99].Improving Oxygenation: CPAP therapy helps improve oxygenation by ensuring a continuous flow of oxygen-rich air into the lungs throughout the night. By preventing episodes of hypoxemia (low oxygen levels in the blood) associated with OSA, CPAP therapy reduces the strain on the cardiovascular system and minimizes the risk of cardiovascular complications.Alleviating Symptoms: CPAP therapy effectively alleviates the symptoms of OSA, including excessive daytime sleepiness, fatigue, morning headaches, and cognitive impairment. By increasing restorative sleep and reducing sleep fragmentation, CPAP therapy helps individuals with OSA feel more refreshed and alert during the day, improving overall quality of life.Reducing Cardiovascular Risk: Effective management of OSA with CPAP therapy has been shown to reduce the risk of cardiovascular events [100,101]. By improving oxygenation, CPAP therapy helps mitigate the cardiovascular consequences of OSA and improve long-term cardiovascular outcomes.

### 7.2. Oral Appliances Used in the Treatment of Sleep Apnea

Several non-invasive oral devices exist for use in patients with mild to moderate OSA. The most common designs are mandibular advancement devices (MADs). These custom-fit oral appliances are used during sleep to reposition the mandible forward, thereby enlarging the upper airway and preventing its collapse or obstruction during sleep. Other devices include palate lift devices and tongue retention devices. These devices are briefly described below:MAD consists of an upper and lower dental tray connected by adjustable metal or plastic hinges. The appliance is custom-made to fit the individual’s teeth and oral anatomy. By adjusting the hinges, the lower jaw can be gradually advanced relative to the upper jaw, which helps open and stabilize the upper airway, thereby preventing obstruction and increasing airway patency [102]. Physiologically, these devices work to enlarge the oropharynx and velopharynx during sleep and activate stretch receptors, reducing airway collapse. While effective for mild to moderate OSA, it is ineffective in severe cases. Negative side effects are also reported, including jaw pain, tenderness, and hypersalivation [103].Palate lift devices: These devices are designed to lift and stabilize the soft palate, thereby enlarging the upper airway and reducing the risk of airway collapse during sleep. They consist of a custom-fitted acrylic or silicone appliance that is inserted into the mouth and positioned against the palate. The device incorporates a mechanism, such as a spring-loaded component or an adjustable screw, that applies upward pressure to the soft palate, lifting it and preventing its collapse during sleep [104].Tongue retention devices: These devices are relatively older and becoming outdated and obsolete due to newer innovations in MADs [105]. Tongue retention devices primarily focus on retaining the tongue in a forward position to prevent its collapse and upper airway obstruction during sleep.

### 7.3. Surgical Treatment of Sleep Apnea

Uvulopalatopharyngoplasty (UPPP)

UPPP is the most common surgical procedure used in the treatment of OSA. Indications include an AHI greater than 15, oxygen desaturation less than 90%, and significant cardiac abnormalities associated with OSA [106]. It aims to alleviate airway obstruction by removing excess tissue from the throat and palate, including the uvula, soft palate, and pharyngeal walls, thereby widening the upper airway and reducing the risk of collapse during sleep. UPPP may effectively reduce the severity of OSA and alleviate symptoms such as snoring and daytime sleepiness in some patients, particularly those with obstruction at the soft palate and uvula levels. Studies have shown a significant improvement in AHI with UPPP when compared to observational controls [83,107]. Furthermore, oxygenation status after UPP has been shown to improve significantly. However, the success rate of UPPP can vary, and not all patients experience significant improvement in symptoms. Additionally, UPPP may be less effective for individuals with severe OSA or those with multilevel airway obstruction [108]. A scoring system known as the Friedman staging system was created in order to categorize and score patients into three stages based on palate position, tonsil size, and BMI. This scoring system may help determine patients who are likely to benefit from UPP and situations in which surgery may not be effective [109]. It should be noted that surgery is usually considered a last resort option for OSA when patients express significant frustration and discomfort from CPAP or oral appliances.

2.Hypoglossal Nerve Stimulation

Hypoglossal nerve stimulation is a novel and promising treatment option for OSA in individuals who have not responded well to traditional therapies. This innovative approach involves the implantation of a stimulating electrode that delivers electrical stimulation to the medial branch of the hypoglossal nerve in order to enhance tongue movement. By stimulating the hypoglossal nerve during sleep, the device helps to prevent the collapse of the upper airway by promoting the forward movement of the tongue and opening of the throat. A respiration sensor is placed between the internal and external intercostal muscles to detect inspiratory power. A pulse generator is placed within the chest wall in order to trigger the hypoglossal nerve electrodes in response to the initiation of respiration. Collectively, this system times stimulation of the hypoglossal nerve with inspiratory effort during sleep [110,111]. This mechanism of action targets one of the key anatomical contributors to OSA, namely the collapse of the tongue and soft tissues at the back of the throat during sleep. Randomized control trials have been promising thus far, showing a reduction in AHI of nearly 70% over 12 months of use, with no serious adverse outcomes [111,112].

### 7.4. Lifestyle Modifications

Obesity is a significant public health challenge in the modern day and likely contributes to the drastic rise in the incidence of sleep apnea over the last two decades. Narrowing of the upper respiratory muscles occurs because of the accumulation of fatty tissue, leading to obstruction. Obesity is also an independent risk factor for many OSA-related complications, such as stroke, hypertension, and coronary artery disease [95]. Weight loss has been shown to significantly improve symptoms and lifestyle in patients with OSA. Diet modifications, regular exercise, and medications can help achieve significant weight loss. If these modifications are unsuccessful and CPAP therapy does not achieve the desired improvement, surgical procedures can be considered. For example, bariatric surgery has been shown to cause significant weight loss and improvement in metabolic derangements in patients with OSA [8,113,114].

Smoking, which doubles the likelihood of OSA [115], and alcohol, which estimates an increased risk by 25% [17], are also modifiable risk factors that can decrease the severity of OSA [116].

### 7.5. Treatment Outcomes

#### 7.5.1. Atrial Fibrillation

CPAP compliance has been shown to reduce atrial fibrillation recurrence. Consistent compliance with CPAP therapy is especially important in patients with atrial fibrillation because conventional therapies, such as cardioversion and radiofrequency ablation, are far less effective in patients with OSA than in the general population. Untreated OSA patients undergoing cardioversion have a risk of recurrence of 82% within 1 year, nearly double the rate of patients undergoing cardioversion who were compliant with CPAP [100]. Additionally, several studies showed that patients undergoing radiofrequency ablation for atrial fibrillation had significantly higher rates of recurrence when compared to patients who were CPAP compliant [117,118]. CPAP was also shown to help reverse left atrial volumetric abnormalities with just 12 weeks of compliance and improved left atrial remodeling over 24 weeks [117]. Due to a coexisting propensity for bradyarrhythmias during sleep, providers should also be careful prescribing traditional rate control medications such as beta blockers to patients with OSA.

#### 7.5.2. Cardiac Remodeling and Heart Failure

By reducing the severity of autonomic dysregulation, intrathoracic pressure variations, and inflammatory signaling, CPAP therapy prevents cardiac remodeling and may help slow the progression of OSA-induced heart failure. Several studies have specifically demonstrated a reduction in cardiac remodeling with CPAP therapy [119,120]. Physiologically, CPAP therapy significantly reduces the large fluctuations in intrathoracic pressure, transmural pressure, and wall tension across the RV and LV. Consequently, it causes afterload reduction and myocardial oxygen demand, which over time, will prevent compensatory cardiac remodeling, which is regularly seen in untreated sleep apnea. By reducing the frequency and duration of apneic episodes, CPAP also reduces hypoxia and hypercapnia-induced oxidative stress, inflammation, and sympathetic nervous system activation [121].

#### 7.5.3. Systemic Hypertension

Numerous primary research studies and meta-analyses have demonstrated the clear benefit of CPAP therapy in reducing systolic and diastolic blood pressure over the long term [122]. The greatest reductions in systemic blood pressure are observed in patients with moderate–severe OSA by AHI [123]. Differences in compliance with CPAP therapy are also highly likely to influence outcomes [124]. Additionally, traditional anti-hypertensive medications are usually effective, albeit less effective than in the general population. Counseling on diet, exercise, and weight loss are also important factors for all patients with hypertension and OSA. Other options for blood pressure reduction in OSA patients include surgical modifications such as uvulopalatopharyngoplasty, tongue base surgery, upper airway stimulation, or hypoglossal nerve stimulation. Recent meta-analyses have demonstrated a significant reduction in BP after sleep surgery. Despite this, CPAP, followed by medical management, remains the standard of care due to lack of invasiveness, ease of use, and cost-effectiveness [125].

#### 7.5.4. Pulmonary Hypertension

Long-term CPAP therapy reduces the duration and frequency of apneic episodes via the maintenance of airway patency and the prevention of collapse. This helps prevent autonomic dysregulation and fluctuations in negative intrathoracic pressure. Consequently, CPAP therapy may help avoid irreversible pulmonary vascular and RV structural remodeling. Long-term CPAP use has been associated with a reduction in daytime pulmonary artery systolic pressure [126].

### 7.6. Future Directions

OSA is a well-studied condition with known cardiovascular sequelae. Efforts focused on treatment to reduce the severity of OSA, and, in turn, morbidity of OSA will be the mainstay of future studies. Currently, CPAP is the mainstay of treatment and has been shown to improve the cardiovascular consequences of OSA [117,118,119,120,121,122,123,124,125,126]. Though studies have shown improvement in OSA severity, improvement in cardiovascular effects has not been consistently shown for other treatments. Additionally, pharmacologic therapy remains an emerging field of research for the treatment of OSA. Medical therapy with eszopiclone, zolpidem, acetazolamide, a combination of oxybutynin/atomexetine, or aminopyridine has shown promising improvements, though they lack the long-term clinical trials to investigate its effect on OSA and sequelae [76].

Treatment compliance with oral appliances, surgical treatment, and medical therapy provide hopeful effects in OSA, and long-term studies focusing on these therapies may uncover evidence of effective CPAP alternatives in the near future.

## 8. Conclusions

OSA poses a significant burden on cardiovascular health, with mounting evidence linking it to a spectrum of cardiovascular abnormalities and adverse events. Through epidemiological studies, clinical observations, and mechanistic research, it has become increasingly evident that OSA is not merely a respiratory disorder but a complex condition with far-reaching implications for cardiovascular morbidity and mortality. The association between OSA and cardiovascular disease underscores the importance of comprehensive evaluation and management strategies aimed at addressing both the respiratory and cardiovascular aspects of this condition. By elucidating the underlying pathophysiological mechanisms and exploring therapeutic interventions targeting OSA-related cardiovascular complications, clinicians and researchers can strive to improve outcomes and enhance the quality of life for individuals affected by this challenging disorder. Moving forward, continued research efforts are needed further to unravel the intricate relationship between OSA and cardiovascular health, paving the way for innovative approaches to prevention, diagnosis, and treatment in this field.

## Figures and Tables

**Table 1 jcm-13-03223-t001:** Apnea–Hypopnea Index score. A higher score is indicative of more severe sleep apnea.

Apnea–Hypopnea Index (AHI) Score (Apnea or Hypopnea Episodes/Hour)	OSA Severity
<5	Normal (no sleep apnea)
5–15	Mild sleep apnea
15–30	Moderate sleep apnea
>30	Severe sleep apnea

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
