# Peer review of "The Effects of Obstructive Sleep Apnea on the Cardiovascular System: A Comprehensive Review"

_jcm, 2024, doi:10.3390/jcm13113223_

Round 1

Reviewer 1 Report

Comments and Suggestions for Authors

Comments to the authors

Thank you for inviting me to review the manuscript entitled “The effects of OSA on the Cardiovascular System: a comprehensive review”.

This is a literature review on the topic of OSA. The article is clearly written. Although it does not add anything more to the topic itself, it surely provides a comprehensive view and summary of mechanisms of actions explained in an accessible and easy way; risk factors; epidemiology, treatments, among others. 

A major problem of this paper is that it is a very long manuscript. It is more similar to a book than to a scientific paper. I suggest that the authors summarize major concepts in diagram and tables (see below for specific comments). Also, I suggest that the authors merge the epidemiology section within the introduction, as their focus should be on CV consequence and etiopathogenesis of OSA increasing CV conditions. This could be a way at least to shorten the text. Also, the authors could utilize numbers to divide section and to facilitate the reader to follow through the structure of the paper. 

Specific comments are below:

Even if this is a literature review without any specific search strategy, the authors should still include a brief method section where they present search terms, consulted databases, and question aimed to be answered. 

I suggest that the authors add a figure or a diagram to summarize the mechanisms of action discussed in pages 3-4. 

HSAT vs PSG: the authors should include those comorbidities / scenarios that constitute an indication for PSG vs HSAT. 

Other diagnostic criteria utilizing AHI or RDI etc should be mentioned and discussed. 

It may be worth it explaining why diagnostic criteria based on AHI have been challenged  in recent years (page 5, line 189).

Reference 40, 41 at page 6. Are the authors sure about the causative factor of OSA inducing AF? Or are they association studies?  

In all the CV conditions listed, it would be valuable to also indicate the prevalence of the specific conditions among the general population, so that it is at still qualitative evident that there is a difference with their prevalence among OSA individuals. Otherwise, the mere indication of the prevalence of such conditions among OSA is left without any comparison. The authors have most likely included these data, but not for all the CV conditions presented. 

Page line 411-419: these are strong statements that need to be carefully cited with robust evidence. Otherwise, the authors should soften their tones (same as lines 512-513). 

To increase the readability of the text, I suggest that the authors summarize their content into a table. For example, it would be nice and of immediate visualization for the reader if a table could show specific CV conditions consequence of untreated OSA, with prevalence in OSA patients vs prevalence in general population, and pathogenesis of such condition as a result of OSA. Basically, what they discussed in writing. However, the writing is very long and summary lacks to this article. 

Not clear how “Obesity” fits into the third part of the article on treatment mechanisms and evidence. Or they create a “lifestyle modification” and they merge this information, or just delete the passage. 

Also the treatment here provided should be introduced in the table suggested earlier, where the authors can summarize at what level of the pathogenesis or what specific CV condition these treatment prove to be beneficial. 

Author Response

Our team would like to thank the Reviewers and Editors for their feedback and insightful suggestions, which have substantially improved the manuscript. We have addressed each issue raised by the Reviewers in the point-by-point response outlined below. Comments are listed in italics with responses in blue. All page numbers refer to their locations in the revised manuscript and can be found in the revised manuscript file highlights in yellow.

Thank you for inviting me to review the manuscript entitled “The effects of OSA on the Cardiovascular System: a comprehensive review”.

This is a literature review on the topic of OSA. The article is clearly written. Although it does not add anything more to the topic itself, it surely provides a comprehensive view and summary of mechanisms of actions explained in an accessible and easy way; risk factors; epidemiology, treatments, among others. 

A major problem of this paper is that it is a very long manuscript. It is more similar to a book than to a scientific paper. I suggest that the authors summarize major concepts in diagram and tables (see below for specific comments).

Response: The length of the manuscript is acknowledged, and significant reductions have been made. The text which has been cut from the manuscript can be seen as “deleted text” as tracked changes.

Also, I suggest that the authors merge the epidemiology section within the introduction, as their focus should be on CV consequence and etiopathogenesis of OSA increasing CV conditions. This could be a way at least to shorten the text.

Response: The introduction and epidemiology sections have been merged, and the previous epidemiology section has been deleted. Please see the following addition to the introduction which can be found on page 2:

“A multi-ethnic study examining sleep-disordered breathing found a prevalence of OSA to be 62.9% of white individuals, 63.5% in African American individuals, 71.5% of Hispanic individuals, and 66.4% of Chinese individuals. Furthermore, many studies have shown an increased prevalence of OSA in males compared to females. One study estimated the prevalence to be two-fold greater in men. Additionally, untreated severe OSA in women were noted to have increased cardiovascular mortality as compared to men.”

Also, the authors could utilize numbers to divide section and to facilitate the reader to follow through the structure of the paper. 

Specific comments are below:

Even if this is a literature review without any specific search strategy, the authors should still include a brief method section where they present search terms, consulted databases, and question aimed to be answered. 

Response: Thank you for the suggestion. We have included a brief “Methods” Section which begins on page 2 after the introduction and extends into the first paragraph of page 3. See below:

“Methods

Bibliographic databases, including Pubmed and Embase were used for this literature review to answer the question, “What are the cardiovascular consequences of obstructive sleep apnea?” Keywords used in this search included obstructive sleep apnea, OSA, sleep-disordered breathing, cardiovascular effects, cardiac effects, cardiovascular consequences, and cardiac consequences. Article types included observational studies, epidemiological studies, retrospective studies, randomized controlled trials, systematic reviews, and meta-analyses. Several articles were obtained from previously referenced publications.”

I suggest that the authors add a figure or a diagram to summarize the mechanisms of action discussed in pages 3-4. 

Response: Thank you for the suggestion. We have addended Figure 1 which now provides a thorough visual representation of the mechanisms of action discussed in the “Mechanisms of Action of OSA” section. Please see the revised Figure 1 which can be located after the references section.

HSAT vs PSG: the authors should include those comorbidities / scenarios that constitute an indication for PSG vs HSAT. 

Other diagnostic criteria utilizing AHI or RDI etc should be mentioned and discussed. 

It may be worth it explaining why diagnostic criteria based on AHI have been challenged in recent years (page 5, line 189).

Response: Thank you for the suggestion. The authors have added specific comorbidities which constitute the use of PSG instead of HSAT, which can be found in the last paragraph on page 5:

“Additionally, patients with comorbidities including significant cardiac or respiratory disease, neuromuscular conditions that may affect respiratory muscle strength, chronic opioid use, awake hypoventilation, history of stroke, or severe insomnia are not recommended to utilize HSAT and should undergo PSG instead.”

Reference 40, 41 at page 6. Are the authors sure about the causative factor of OSA inducing AF? Or are they association studies?  

Response: Thank you for pointing this out. We have addended this section of the manuscript which can be found on page 6-7, beginning in the final paragraph of page 6. It now reads as follows:

“Growing evidence also implicates a concomitant prevalence of AF of 21-74% in patients with OSA, which may suggest that OSA may play a role in AF pathogenesis and development and may also trigger the persistence of AF.40,41

In all the CV conditions listed, it would be valuable to also indicate the prevalence of the specific conditions among the general population, so that it is at still qualitative evident that there is a difference with their prevalence among OSA individuals. Otherwise, the mere indication of the prevalence of such conditions among OSA is left without any comparison. The authors have most likely included these data, but not for all the CV conditions presented. 

Response: Thank you for the suggestions. We have incorporated data regarding the prevalence of certain conditions which can be found in the updated manuscript as highlighted text on page 7, 8, and 9. A table was created which outlines the prevalence of specific conditions among the general population and patients with OSA. Table 2 can be found at the bottom of the manuscript after the references section.

Page line 411-419: these are strong statements that need to be carefully cited with robust evidence. Otherwise, the authors should soften their tones (same as lines 512-513). 

Response: Thank you for the suggestion. The tone of the paragraph on sudden cardiac death has been softened and now includes a reference documented the increased risk of sudden cardiac death and OSA patients. The changes are highlights in the paragraph on sudden cardiac death which can be found on page 11 and 12.

Response: Thank you for the suggestion. The section on cerebrovascular disease and stroke has been updated, and the tone has been changed to reflect a more nuanced approach. An additional reference has been included after the first sentence:

“OSA may be an independent risk factor for ischemic stroke. In OSA patients, significant contributing factors include cerebrovascular disease, atrial fibrillation, and sleep fragmentation.”

To increase the readability of the text, I suggest that the authors summarize their content into a table. For example, it would be nice and of immediate visualization for the reader if a table could show specific CV conditions consequence of untreated OSA, with prevalence in OSA patients vs prevalence in general population, and pathogenesis of such condition as a result of OSA. Basically, what they discussed in writing. However, the writing is very long and summary lacks to this article. 

Response: Thank you for the suggestion. A new table was created which outlines data regarding the cardiovascular effects of OSA, including prevalence of comorbidities in the general population, prevalence in OSA patients, and pathophysiology. Table 2 can be found at the bottom of the manuscript after the references section.

Not clear how “Obesity” fits into the third part of the article on treatment mechanisms and evidence. Or they create a “lifestyle modification” and they merge this information, or just delete the passage. 

Response: Thank you for the suggestions. We have appropriately revised this paragraph and titled it “Lifestyle Modifications” and have added additional commentary and references. Please see below, which can be found on page 17 and 18:

“Lifestyle Modifications

Obesity is a significant public health challenge in the modern day, and likely contributes to the drastic rise in the incidence of sleep apnea over the last two decades. Narrowing of the upper respiratory muscles occurs because of the accumulation of fatty tissue, leading to obstruction. Obesity is also an independent risk factor for many OSA-related complications, such as stroke, hypertension, and coronary artery disease.99 Weight loss has been shown to significantly improve symptoms and lifestyle in patients with OSA. Diet modifications, regular exercise, and medications can help achieve significant weight loss. If these modifications are unsuccessful and CPAP therapy does not achieve the desired improvement, surgical procedures can be considered. For example, bariatric surgery has been shown to cause significant weight loss and improvement in metabolic derangements in patients with OSA.125,126

Smoking, which doubles the likelihood of OSA,19 and alcohol, which estimates an increased risk by 25%,20 are also modifiable risk factors that can decrease the severity of OSA.”

Also the treatment here provided should be introduced in the table suggested earlier, where the authors can summarize at what level of the pathogenesis or what specific CV condition these treatment prove to be beneficial. 

Response: Thank you for the suggestion. The treatment section outlines the treatment for OSA which is primarily CPAP in addition to several other options for patients who are CPAP intolerant. The authors feel that the topic of treatment is adequately addressed in the paper.

Reviewer 2 Report

Comments and Suggestions for Authors

The relationship between sleep apnea and various cardiac disorders is very interesting considering the high prevalence of OSAS worldwide. Despite this, the article adequately summarizes and clearly presents the relationship between obstructive sleep apnea syndrome (OSAS) and every related cardiac pathologies.
The paper contains minor text formatting errors that need to be corrected.
Additionally, it would be beneficial to develop a table or diagram summarizing the pathophysiological correlation between OSAS and various cardiac disorders, to make the causal link between OSAS and these disorders more immediate and easier to understand.
In order to empower the section regarding atrial fibrillation I strongly suggest to include “
Incidence and Determinants of Spontaneous Cardioversion of Early Onset Symptomatic Atrial Fibrillation. Medicina (Kaunas). 2022 Oct 24;58(11):1513. doi: 10.3390/medicina58111513. PMID: 36363470; PMCID: PMC9693621.” As it resumes the main determinants of AF.
Moreover, in order to give the paper a more clinical setting, It would be useful to include a commentary from the author, based on their experience, regarding the management of these patients, especially in relation to handling patients with multiple comorbidities.

In order to make your review more complete you should add a “future directions” section considering the actual gaps in medical assistance

I believe that the article could be accepted after following all of my suggestions that consist in minor revisions.

Comments on the Quality of English Language

moderate english corrections are required

Author Response

Our team would like to thank the Reviewers and Editors for their feedback and insightful suggestions, which have substantially improved the manuscript. We have addressed each issue raised by the Reviewers in the point-by-point response outlined below. Comments are listed in italics with responses in blue. All page numbers refer to their locations in the revised manuscript and can be found in the revised manuscript file highlights in yellow.

Reviewer 1:

The relationship between sleep apnea and various cardiac disorders is very interesting considering the high prevalence of OSAS worldwide. Despite this, the article adequately summarizes and clearly presents the relationship between obstructive sleep apnea syndrome (OSAS) and every related cardiac pathologies.The paper contains minor text formatting errors that need to be corrected. Additionally, it would be beneficial to develop a table or diagram summarizing the pathophysiological correlation between OSAS and various cardiac disorders, to make the causal link between OSAS and these disorders more immediate and easier to understand.

 Response: Thank you for the suggestion. We have created a table which appropriately summarizes the pathophysiologic correlation between OSA and cardiovascular disorders with which it is associated. Table 2 can be found after the references section at the bottom of the manuscript.

In order to empower the section regarding atrial fibrillation I strongly suggest to include “Incidence and Determinants of Spontaneous Cardioversion of Early Onset Symptomatic Atrial Fibrillation. Medicina (Kaunas). 2022 Oct 24;58(11):1513. doi: 10.3390/medicina58111513. PMID: 36363470; PMCID: PMC9693621.” As it resumes the main determinants of AF.
Moreover, in order to give the paper a more clinical setting, It would be useful to include a commentary from the author, based on their experience, regarding the management of these patients, especially in relation to handling patients with multiple comorbidities.

 In order to make your review more complete you should add a “future directions” section considering the actual gaps in medical assistance.

 I believe that the article could be accepted after following all of my suggestions that consist in minor revisions.

Round 2

Reviewer 1 Report

Comments and Suggestions for Authors

The authors have now provided a revised version where they sufficiently addressed previously expressed concerns. Figure 1 is particularly important for the understanding of etiopathophysiolgy. Table 2 is also very well summarized and readily conveys the message of an increased predisposition within patients with OSA to those listed comorbidities. I am satisfied with the current version.